# SARS-CoV-2 infection induces protective immunity and limits transmission in Syrian hamsters

Prabhuanand Selvaraj[1],*, Christopher Z Lien[1],*⑩, Shufeng Liu[1], Charles B Stauft[1], Ivette A Nunez[1], Mario Hernandez[2], Eric Nimako[2], Mario A Ortega[2], Matthew F Starost[3], John U Dennis[2] ⑩, Tony T Wang[1] ⑩

A critical question in understanding the immunity to SARS-COV-2 is whether recovered patients are protected against re-challenge and transmission upon second exposure. We developed a Syrian hamster model in which intranasal inoculation of just 100 $TCID_{50}$ virus caused viral pneumonia. Aged hamsters developed more severe disease and even succumbed to SARS-CoV-2 infection, representing the first lethal model using genetically unmodified laboratory animals. After initial viral clearance, the hamsters were re-challenged with $10^5$ $TCID_{50}$ SARS-CoV-2 and displayed more than 4 log reduction in median viral loads in both nasal washes and lungs in comparison to primary infections. Most importantly, re-challenged hamsters were unable to transmit virus to naïve hamsters, and this was accompanied by the presence of neutralizing antibodies. Altogether, these results show that SARS-CoV-2 infection induces protective immunity that not only prevents re-exposure but also limits transmission in hamsters. These findings may help guide public health policies and vaccine development and aid evaluation of effective vaccines against SARS-CoV-2.

## Introduction

The outbreak of the severe acute respiratory syndrome coronavirus 2 (SARS-CoV-2) has quickly turned into a global pandemic (1). SARS-CoV-2 is described to have an $R_0$ value of 2–5 (2) and is well adapted to transmission among humans (3 *Preprint*) through aerosol and possibly the fecal–oral route (4). Infectious virus has been isolated from oro- or naso-pharyngeal swabs (5), sputum (5), and stool samples (6) of individuals with severe coronavirus disease 2019 (COVID-19). Limited evidence suggests that recovered individuals are likely to be protected from reinfection (7), which has also been experimentally demonstrated in animal models (8, 9, 10). However, it remains unclear whether postinfection immunity will prevent a recovered individual from becoming an active transmitter if re-exposed. Such information is critical needed for decision-making of public health policies and evaluating efficacy of vaccine candidates. To address this important question, we used a Syrian hamster model to evaluate the protective efficacy of SARS-CoV-2 infection against secondary challenge and transmission.

## Results

### Induction of interstitial pneumonia in Syrian hamsters by 100 $TCID_{50}$ SARS-CoV-2

Syrian hamsters of 7–9 wk old that were intranasally inoculated with $10^5$ $TCID_{50}$ (50% tissue culture infectious dose) SARS-CoV-2 had weight loss for up to 7 d (Fig 1A). Lower inocula (from $10^4$ to 10 $TCID_{50}$) caused lesser or no weight loss (Fig 1A). To accurately describe active viral replication, we measured levels of subgenomic viral mRNA (sgmRNA) by real-time PCR (11) in nasal washes (NW) of hamsters inoculated with $10^5$ $TCID_{50}$ at 2, 3, 5, and 7 days post-inoculation (d.p.i.). We also assessed viral RNA (vRNA) and sgmRNA in lung tissue and organs at 3 and 5 d.p.i. Interestingly, high levels of sgmRNA were observed in NW with a median peak of 4.87 (range 3.73–5.38) $\log_{10}$ sgmRNA copies/5 $\mu$l on 2 d.p.i, but the median quickly subsided to below detection limit on 5 d.p.i. (Fig 1B). On 3 d.p.i., high levels of sgmRNA and vRNA could be detected in both NW and in the lungs (Fig 1C and D). On Day 5 p.i., although high vRNA levels were primarily detected in the upper and lower respiratory tract (Fig 1E), the sgmRNA level in respiratory tract (nasal turbinate, trachea, and lung) declined to just above the detection limit, whereas it remained substantial in intestine, spleen and kidney (Fig 1F). Viral pneumonia was also confirmed by HE staining of lung tissues at 2–7 d.p.i (Figs 1G–N and S1). The infected lung was characterized as many multifocal consolidations composed of foamy macrophages, atypical pneumocytes hyperplasia, interstitial lymphocytic, and neutrophilic infiltrates. Scattered areas of airway

---

[1]Division of Viral Products, Center for Biologics Evaluation and Research, US Food and Drug Administration, Silver Spring, MD, USA   [2]Division of Veterinary Services, Center for Biologics Evaluation and Research, US Food and Drug Administration, Silver Spring, MD, USA   [3]Division of Veterinary Resources, Diagnostic and Research Services Branch, National Institutes of Health, Rockville Pike, MD, USA

Correspondence: tony.wang@fda.hhs.gov
*Prabhuanand Selvaraj and Christopher Z Lien contributed equally to this work

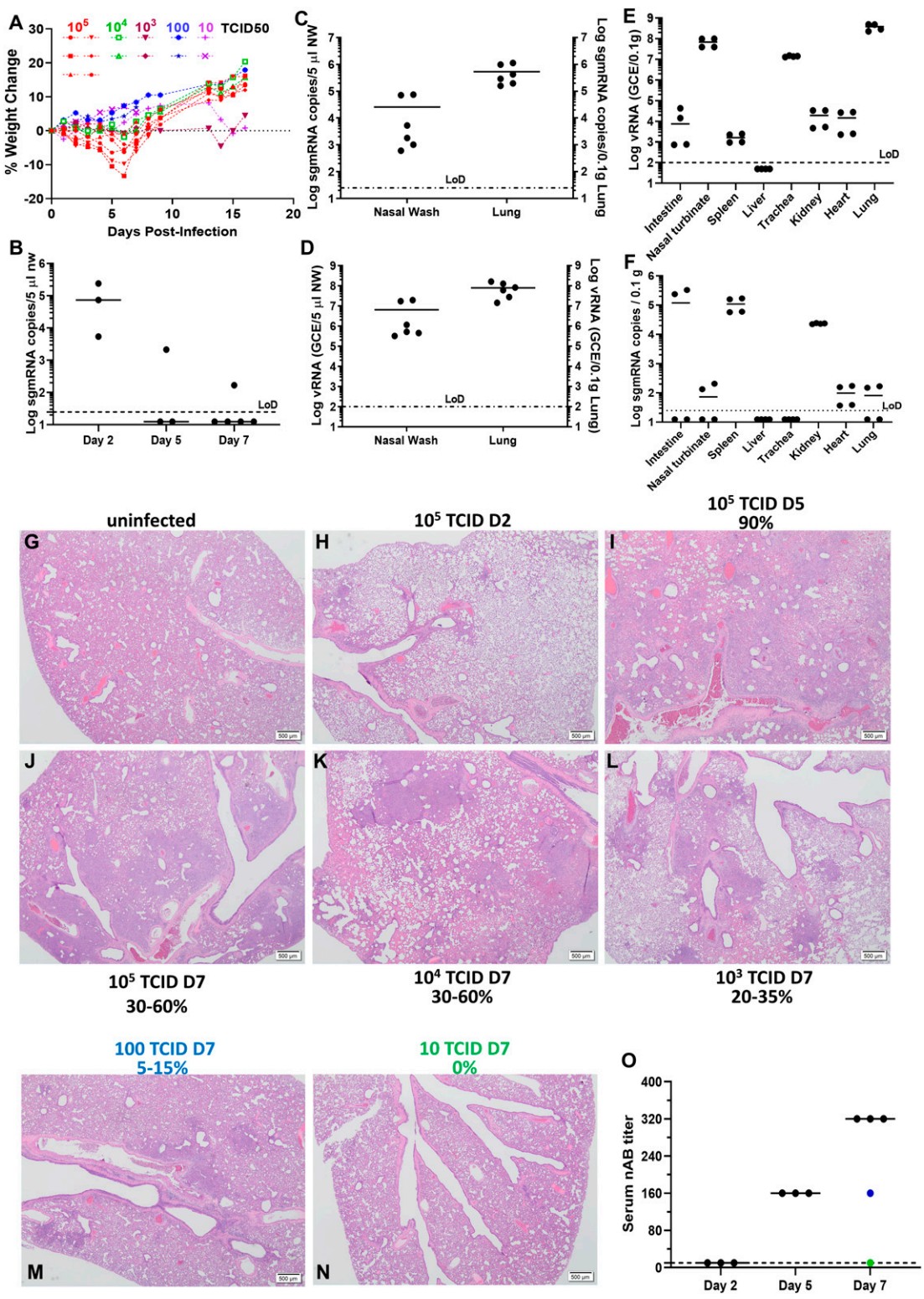

**Figure 1. Viral load and histopathological changes in female Syrian hamsters intranasally challenged with SARS-CoV-2**
.**(A)** Weight change profiles of hamsters that have been intranasally infected by indicated doses of SARS-CoV-2 USA/WA1-2020. **(B)** Subgenomic viral mRNA (sgmRNA) detected in nasal washes of SARS-CoV-2 challenged hamsters (N = 3) on 2, 5, 7 dpi. **(C, D)** SgmRNA and viral RNA (vRNA) (D) detected in nasal washes and the lungs of SARS-CoV-2–challenged hamsters (N = 6) on 3 dpi. **(E, F)** vRNA and sgmRNA detected in various organs of SARS-CoV-2–challenged hamsters (N = 4) on 5 dpi. The detection limit (25 copies for sgmRNA and 100 copies for vRNA per reaction) was shown with the dotted line. **(G, H, I, J)** Hematoxylin and eosin (H&E) staining of the lungs of $10^5$ TCID$_{50}$ SARS-CoV-2–challenged hamsters on 2, 5, and 7 dpi. **(K, L, M, N)** Lung tissues from hamsters challenged with indicated doses of virus were estimated for the percentage of consolidated areas. Scale bar in (G, H, I, J, K, L, M, N) = 500 $\mu$m. **(O)** Neutralizing antibody titers of each hamster at the end of the study. Blue dot denotes the hamster that received 100 TCID$_{50}$; Green dot represents the hamster that received 10 TCID$_{50}$.

proteinaceous fluid were seen (Fig S1). Notably, the percentage of consolidated areas in the lung directly correlated with the amount of the initial inoculum. Shown in Fig 1M, a viral inoculum of as low as 100 $TCID_{50}$ was able to cause 10–15% consolidated areas in the lung. By contrast, inoculation of 10 $TCID_{50}$ did not result in observable viral pneumonia in one out of two hamsters. Serum-neutralizing antibody (nAB) titers rose from Day 2 to 5 dpi with mean nAB titers of 160 to 320 at days 5 and 7, respectively (Fig 1O).

On day 2 p.i., numerous individual necrotic respiratory epithelial cells were seen in the middle nasal cavity (Fig S1A–C). On Day 5 p.i., the nasal cavity showed neutrophilic infiltrate, edema, degeneration and loss of submucosal glands and nerves. Olfactory mucosal cells in the nasal cavity were abnormally arranged. Loss of cilia and edema of submucosa was also noticed. In the lung, only a few isolated neutrophilic infiltrates and mild type II pneumocyte hyperplasia were observed on Day 2 p.i. without obvious viral pneumonia (Fig S1H and I). On Day 5 & 7 p.i., 30–60% areas in the lung were consolidated and filled with abundant atypical and lesser typical pneumocyte and terminal bronchiolar epithelial hyperplasia and hypertrophy (Fig S1J–L). Airways in these foci had mild to moderate neutrophilic, lymphocytic, and foamy macrophage infiltrates. In infected trachea, mild mucosal hyperplasia and attenuation with loss of cilia were observed (Fig S1D–G). Lesions were not found in brain including olfactory bulbs (Fig S1M–R), heart, liver, kidney, small intestine, spleen, and salivary gland.

As a model for studying SARS-CoV-2 pathogenesis, Syrian hamsters can also be easily infected through close contact with another infected hamster. As shown in Fig S2, when a naïve hamster, termed as contact hamster, was intranasally given MEM and subsequently housed in the same clean cage with a hamster that was intranasally inoculated with $10^4$ $TCID_{50}$ SARS-CoV-2 (defined as the transmitter), despite no observed weight loss (Fig S2A), the contact hamster developed viral pneumonia (Fig S2C–H), and became seroconverted (Fig S2B). The presence of virus-infected cells in the lung was confirmed by detecting SARS-CoV-2 genomic RNA using RNAscope (Fig S2I–J). This is consistent with previous reports (12).

## Lethal infection of aged Syrian hamsters

Older people have been disproportionally affected by COVID-19, with markedly higher death rate (13). To explore whether this phenomenon is recapitulated in hamsters, we first intranasally inoculated four aged female hamsters (10, 13, and 20 mo old) with $10^5$ $TCID_{50}$ SARS-CoV-2. Two 10-mo-old female hamsters were intranasally inoculated with plain medium as negative controls. Aged hamsters showed marked weight loss (around 20%) within 7 d (Fig 2A) and outward signs of sickness such as sneezing, shortness of breath, shivering, and lethargy (Fig 2B and C). The illness was especially obvious when breathing rates were analyzed, in which the infected aged hamsters were breathing two to three times faster than uninfected animals (Fig 2C). Sustainable levels of subgenomic viral mRNA (sgmRNA) were detected from NW up to 7 d postinfection, indicating a prolonged period of active virus replication in the nose (Fig 2D). The oldest hamster (20 mo old) died on Day 6 p.i. Postmortem histopathology examination indicated that the animal had severe, diffuse bronchointerstitial pneumonia along with myocardial disease and thrombus formation in the left atrium leading to death (Fig 2E). Worth mentioning is that there were airway infiltrates of macrophages and

neutrophils, and necrosis of terminal bronchiolar epithelial cells (Fig 2F). Massive amount of viral genomic RNA was detected in the lung of the deceased hamster (Fig 2G and H). Last, infection of aged male hamsters (13 mo old) resulted in 100% death (n = 4) between day 5 and 8 post-challenge (Fig 2J). Notably, a full panel necropsy did not find lesions in other vital organs (pancreas, small intestine, large intestine, brain, and kidney). Examination of the infected hamster brains did not show the presence of virus genomic RNA (Fig 2I). This finding contrasts with the reported lethal encephalitis seen in a subset of infected K18-hACE2 mice (14 Preprint, 15, 16, 17). In general, the pathology in aged hamsters resembles what has been observed in patients with severe COVID-19 who usually have pneumonitis and/or acute respiratory distress syndrome with increased pulmonary inflammation, thick mucus secretions in the airways, elevated levels of serum pro-inflammatory cytokines, extensive lung damage, and microthrombosis. Neutrophilia predicts poor outcomes in patients with COVID-19 (18), and the neutrophil-to-lymphocyte ratio is an independent risk factor for severe disease (19). It is unclear whether the deceased hamster also developed cytokine storm, which is frequently seen in severe human COVID-19 patients (20). Post-mortem examination of COVID-19 patients has revealed signs of vascular dysfunction (21), which was also seen in hamsters.

## Protection of reinfection and prevention of transmission by postinfection immunity

To access transmission from reinfected animals, a group of 10 hamsters that were previously infected and recovered were re-inoculated intranasally 4 wk later with $10^5$ $TCID_{50}$ SARS-CoV-2 (Fig 3A). Although we have demonstrated that a hamster inoculated with $10^4$ $TCID_{50}$ virus was able to transmit virus to a naïve hamster, we chose a higher inoculum of $10^5$ $TCID_{50}$ to mimic a more extreme situation. The circulating neutralizing antibody titers of these 10 hamsters range from 80 to 320 (Fig 3B). Re-inoculation of these hamsters did not result in any weight loss (Fig 3C). NWs were collected from at 1, 2, 3, and 7 d after reinfection. For most NW samples, the median levels of sgmRNA in NWs collected at all time points except 1 dpi were below the detection limit and the median sgmRNA level on 1 dpi was about four logs lower than that seen during primary infections (Fig 3D), suggesting a significant reduction in active virus replication in the nose. In consistent with this finding, comparison of the amount of infectious virus in the NW also showed a 3–4 log reduction in the secondary infection (Fig 3E). The sgmRNA in the lungs was also undetectable on both Day 3 and 7 postinfection, although a very low level of vRNA could be detected in the lungs (Fig 3F). Pathology examination of the lungs also showed no viral pneumonia, confirming protection of these hamsters from virus-induced disease (Fig 3G).

To understand the impact of prior infection on the ability of re-exposed hamsters to transmit virus, four re-inoculated hamsters were paired with four naïve hamsters (on Day 1 post–re-inoculation) for 24 h and then with another two aged naïve hamsters on Day 2 post-re-inoculation for 24 h in new clean cages. Over a period of 7 d, the contact hamsters showed no signs of sickness, no weight loss, and no detectable viral sgmRNA or RNA in the nose or in the lungs, and no pathology in the lung (Fig 4A–E). By contrast, high amount of vRNA was detected from two control contact hamsters that were exposed to hamsters which had no prior infection and subsequently challenged with $10^5$ $TCID_{50}$ SARS-CoV-2 (Fig 4C). At the end of the study, none of the contact hamsters exposed to re-inoculated

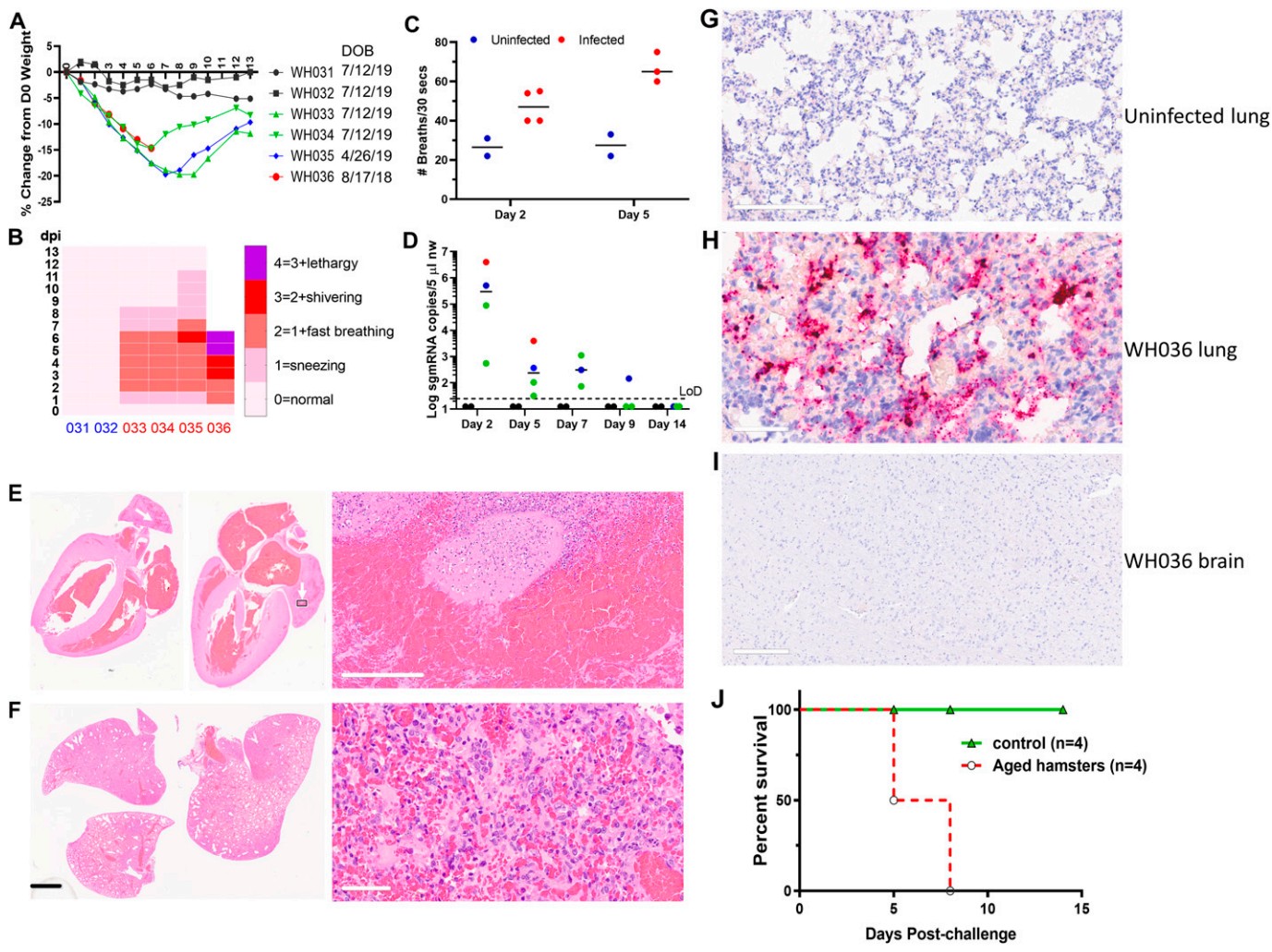

**Figure 2. Prolonged virus replication in nasal cavity of aged Syrian hamsters.**
**(A)** Weight change profile of aged hamsters after intranasal inoculation of $10^5$ TCID$_{50}$ SARS-CoV-2 isolate USA-WA1/2020. The date of birth was indicated to the right of each hamster ID. WH031 and 32 represent negative controls inoculated with media. **(B)** Clinical scores of hamsters. Y-axis represents day post-infection (dpi). Day 0 is the day when inoculum was given. X-axis represents the animal ID with WH031 and 032 being uninfected control. **(C)** Breathing rate was counted by two independent observers and averaged. **(D)** sgmRNA levels in uninfected (black solid circles) and infected hamsters (colored solid circles) on 2,5,7,9,14 dpi. **(A)** Each color corresponds to the same colored animal ID in (A). **(E)** Pathology of the heart of the hamster that died. The thrombus formation was noticed in the left atrium (indicated by white arrows and a small black box) with a closeup image shown to the right. Scale bar = 200 $\mu$m. **(F)** Pathology of the lung of the hamster that succumbed to infection. Scale bar: left = 5 mm; right = 60 $\mu$m. **(G, H, I)** RNAscope images of the uninfected hamster lung (negative), the lung and the brain of WH036. Red dots indicate the presence of viral genomic RNA. **(G, H, J)** Scale bar: 200 $\mu$m in (G), 60 $\mu$m in (H), 300 $\mu$m in (J). **(J)** Survival curves of four aged hamsters.

hamsters developed neutralizing antibodies, implying that they have never been infected (Fig 4F). Altogether, these data suggest that hamsters that have been previously infected are severely impaired in transmitting virus upon re-exposure.

## Discussion

While several studies showing the promise of hamsters, non-human primates, and ferrets to study SARS-CoV-2, disease manifestation in laboratory animals ranges from mild to moderate. Here, we report that aged hamsters develop much more pronounced disease and even fatality upon challenge, recapitulating the

disease manifestation in the older human population. In our study, infected hamsters only occasionally recorded increases in body temperature. Young hamsters, even when losing 15–20% body weight, did not show outward signs of sickness. Aged hamsters, by contrast, exhibited sneezing, shortness of breath, and noticeable lethargy upon infection. Aged hamsters in our study succumbed to infection between Day 5 and 8 p.i. To the best of our knowledge, this is the first lethal model of SARS-CoV-2 using genetically unmodified laboratory animals. The cause of death of aged hamsters is most likely due to severe pneumonia plus thrombus formation in the left atrium. At this time, we are uncertain whether the thrombus formation in the left atrium was directly related to SARS-CoV-2 infection. Future investigation using additional aged hamsters are warranted. Notably, all infected hamsters showed recruitment of

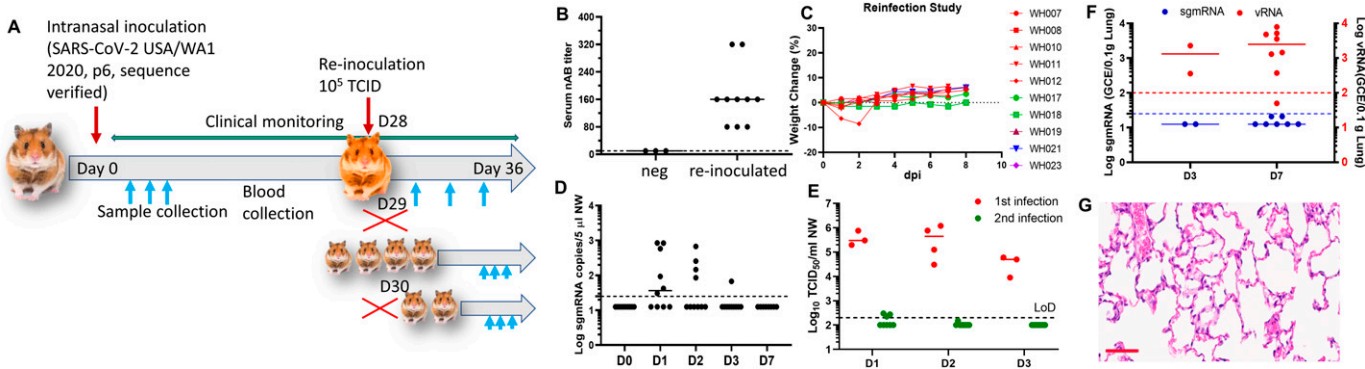

**Figure 3. Post-infection immunity protects re-challenge.**
**(A)** Overall study design of the study. 10 hamsters Fig 1A recovered from infection were reinoculated with $10^5$ $TCID_{50}$ SARS-CoV-2 isolate USA-WA1/2020 on Day 28 post primary infection. **(B)** Neutralizing antibody titers of each hamster at the time of re-inoculation. **(C)** Weight change profile of all re-inoculated hamsters. **(D)** sgmRNA detected from nasal washes collected on 0, 1, 2, 3, and 7 dpi. **(E)** Virus titers of nasal washes collected from first (red) and second (green) infections. **(F)** Viral RNA (red) and sgmRNA levels (blue) in the lungs on 3, 7 dpi. **(G)** A representative HE image of the lung from a re-inoculated hamster. Scale bar = 60 $\mu$m.

both macrophages and neutrophils to the infected lungs. Around 10–15% of COVID-19 patients progresses to acute respiratory distress syndrome triggered by a cytokine storm. It would be interesting to measure other immune correlates in infected aged hamsters, but the relative lack of reagents for conducting immunological assays in hamsters prevented us from measuring cytokine profiles. Nevertheless, aged hamsters offer a superior immunocompetent animal model to study severe form of COVID-19 and will be valuable experimental tools in developing vaccine and public health strategies aimed at ending the global COVID-19.

The most important finding of this study is that prior infection of hamsters induced protective immunity, accompanied by the presence of neutralizing antibodies, in the lung with significantly reduced active virus replication in the nose upon re-exposure and hence restricted transmission. This finding provides important new insights into our understanding of postinfection immunity against SARS-CoV-2. To date, no human reinfections with SARS-CoV-2 have been confirmed; there is also no evidence that clinically recovered individuals continue transmitting SARS-CoV-2 to others. However, it is possible that the virus may replicate in the nose of recovered individuals upon re-exposure such that these persons may facilitate transmission of the virus to the rest of the population. In this scenario, prior infection may have little impact to stop the continuous transmission of the virus. In our study, although active virus replication was sporadically detected from NW of a few re-inoculated hamsters, the overall level of active replication was four logs lower than those in primary infection. We hypothesize the amount of virus shed from these hamsters was so low that the probability of transmitting and infecting another naïve hamster becomes finite. It will be highly desirable for a vaccine to achieve similar effects, that is, to protect against both lower respiratory tract disease and upper respiratory tract disease. In fact, a recent SARS-CoV-2 DNA vaccine conferred protection with significant reductions in median viral loads in bronchoalveolar lavage and nasal swabs in immunized rhesus macaques compared with sham controls (11). Obviously, our study was carried out in hamsters and hence did not address the question of whether SARS-CoV-2 may replicate more efficiently in humans. Nonetheless, our findings support assessing virus replication in the upper respiratory tract, particularly in the nose, in preclinical studies to help with the evaluation of potential vaccine efficacy before clinical trials.

# Materials and Methods

## Viruses and cells

Vero E6 cell line (Cat. no. CRL-1586) was purchased from American Type and Cell Collection and cultured in eagle's MEM supplemented with 10% fetal bovine serum (Invitrogen) and 1% penicillin/streptomycin and L-glutamine. The SARS-CoV-2 isolate USA-WA1/2020 was obtained from Biodefense and Emerging Infectious Research Resources Repository (BEI Resources), National Institute of Allergy and Infectious Diseases (NIAID), National Institutes of Health (NIH), and had been passed three times on Vero cells and one time on Vero E6 cells before acquisition. It was further passed once on Vero E6 cells in our laboratory. The virus has been sequenced verified to contain no mutation to its original seed virus.

## Virus titration

SARS-CoV-2 isolate USA-WA1/2020 was titered using the Reed & Muench Tissue Culture Infectious Dose 50 Assay ($TCID_{50}$/ml) system (22). Vero cells were plated the day before infection into 96 well plates at $1.5 \times 10^4$ cells/well. On the day of the experiment, serial dilutions of virus were made in media and a total of six to eight wells were infected with each serial dilution of the virus. After 48 h incubation, cells were fixed in 4% PFA followed by staining with 0.1% crystal violet. The $TCID_{50}$ was then calculated using the formula: $\log(TCID_{50}) = \log(do) + \log(R)(f + 1)$. Where do represents the dilution giving a positive well, f is a number derived from the number of positive wells calculated by a moving average, and R is the dilution factor.

## SARS-CoV-2 pseudovirus production and neutralization assay

Human codon-optimized cDNA encoding SARS-CoV-2 S glycoprotein (NC_045512) was synthesized by GenScript and cloned into eukaryotic cell expression vector pcDNA 3.1 between the BamHI and XhoI sites. Pseudovirions were produced by co-transfection of Lenti-X 293T cells with psPAX2, pTRIP-luc, and SARS-CoV-2 S expressing plasmid using Lipofectamine 3000. The supernatants were harvested at 48 and 72 h post-transfection and filtered through 0.45-mm membranes.

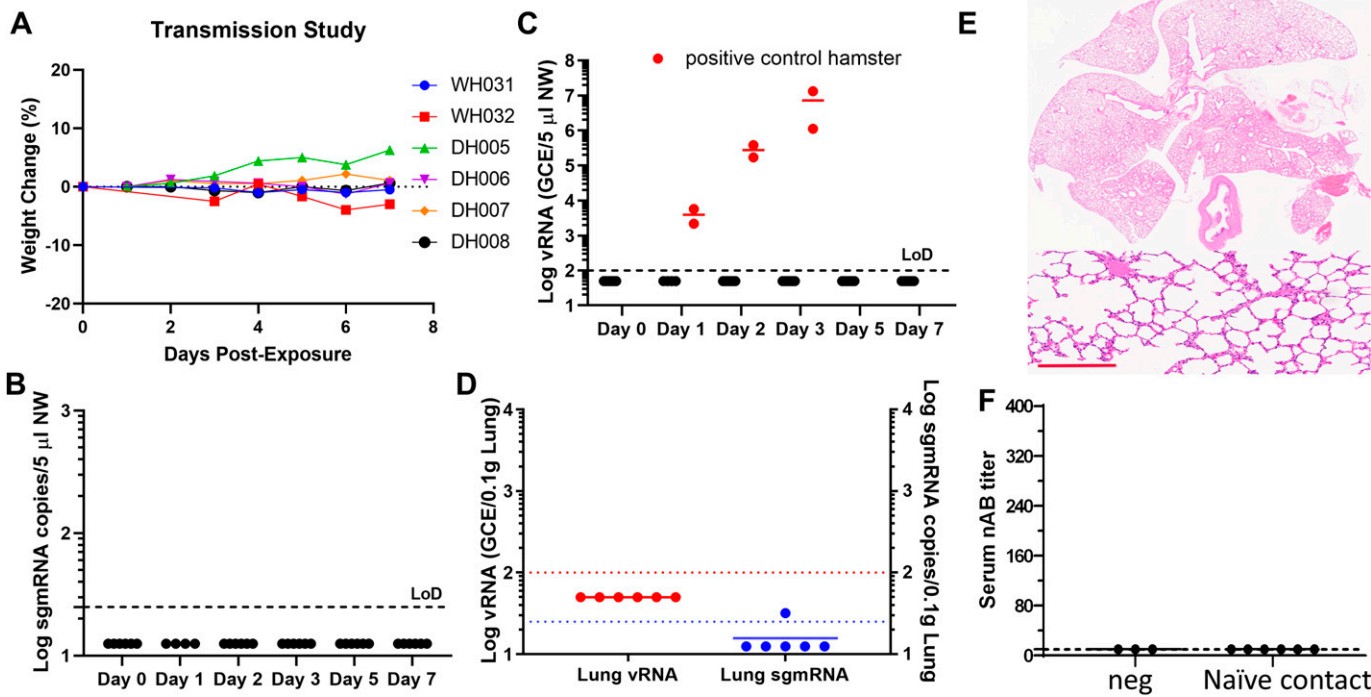

**Figure 4. Post-infection immunity limits transmission.**
The overall study design is depicted in Fig 3A. Four hamsters were paired with four naïve hamsters in clean cages on Day 29 (1 d after re-inoculation) and then another two naïve hamsters on Day 30 (2 d after re-inoculation of the transmitters). Each cage housed two hamsters (one re-inoculated and one naïve). **(A)** Weight change profile of contact hamsters. **(B, C)** sgmRNA and viral RNA (C) in nasal washes collected on 0, 1, 2, 3, 5, and 7 d post-contact. **(C)** Red solid circles in (C) denote two contact hamsters which were exposed to hamsters that were without prior infection but challenged with $10^5$ TCID50 virus. Black solid circles indicate contact those hamsters that were exposed to the re-challenged hamsters. **(D)** Viral loads detected in the lungs. **(E)** Representative HE images of the lung from a contact hamster. Scale bar = 200 $\mu$m. **(F)** Neutralizing antibody titers of each hamster at the end of the study.

For the serum neutralization assay, 50 $\mu$l of SARS-CoV-2 S pseudo-virions were preincubated with an equal volume of medium containing serum at varying dilutions at room temperature for 1 h, then virus–antibody mixtures were added to 293T-hACE2 cells in a 96-well plate. After a 3 h incubation, the inoculum was replaced with fresh medium. Cells were lysed 48 h later, and luciferase activity was measured using luciferin-containing substrate. Controls included cell only control, virus without any antibody control and positive control sera. The end point titers were calculated as the last serum dilution resulting in at least 50% SARS-CoV-2 neutralization. The amount of pseudovirions used in this assay has been determined to give rise to a target input $5 \times 10^5$–$10^7$ RLU/ml, under which condition the neutralization law is observed.

### Hamster challenge experiments

Male and Female outbred Syrian hamsters were purchased from Envigo at 4–5 wk of age. Aged hamsters from the same source were previously purchased and held at The US Food and Drug Administration (FDA) vivarium. All experiments were performed within the biosafety level 3 (BSL-3) suite on the White Oak campus of the U.S. Food and Drug Administration. The animals were implanted subcutaneously with IPTT-300 transponders (BMDS), randomized, and housed two per cage in sealed, individually ventilated rat cages (Allentown). Hamsters were fed irradiated 5P76 (Lab Diet) ad lib, housed on autoclaved aspen chip bedding with reverse osmosis-

treated water provided in bottles, and all animals were acclimatized at the BSL3 facility for 4–6 d or more before the experiments. The study protocol details were approved by the White Oak Consolidated Animal Care and Use Committee and carried out in accordance with the PHS Policy on Humane Care & Use of Laboratory Animals.

For challenge studies, young (5–9 wk old) and aged (10–20 mo old) Syrian hamsters were anesthetized with intraperitoneal injection of anesthetic cocktail (50 mg/kg ketamine + 0.15 mg/kg dexmedetomidine). Intranasal inoculation was performed by pipetting $10^5$ TCID$_{50}$ or desirable doses of SARS-CoV-2 in 50–100 $\mu$l volume dropwise into the nostrils of the hamster under anesthesia. To facilitate recovery, each hamster received 3.3 mg/kg diluted atipamezole solution intraperitoneally immediately after intranasal inoculation. Hamsters were weighed and assessed daily. NW were collected by pipetting ~160 $\mu$l sterile phosphate buffered saline into one nostril when hamsters were anesthetized by 3–5% isoflurane. For tissue collection and blood collection, hamsters were euthanized by intraperitoneal injection of pentobarbital at 200 mg/kg on days 2, 5, 7 as necessary.

### RNA isolation from NW and tissues

RNA was extracted from 50 $\mu$l NW or 0.1-g tissue homogenates using QIAamp vRNA mini kit or the RNeasy 96 kit (QIAGEN) and eluted with 60 $\mu$l of water. 5 $\mu$L RNA was used for each reaction in real-time RT-PCR.

## Real-time PCR assay of SARS-CoV-2 viral and subgenomic RNA

Quantification of SARS-CoV-2 vRNA was conducted using the SARS-CoV-2 (2019-nCoV) CDC qPCR Probe Assay (IDTDNA) using iTaq Universal Probes One-Step Kit (Bio-Rad). The standard curve was generated using 2019-nCoV_N_Positive Control (IDTDNA). The detection limit of the vRNA was determined to be 100 copies/reaction. Quantification of SARS-CoV-2 *E* gene subgenomic mRNA (sgmRNA) was conducted using Luna Universal Probe One-Step RT-qPCR Kit (New England Biolabs) on a Step One Plus Real-Time PCR system (Applied Biosystems). The primer and probe sequences were: SARS2EF: CGATCTCTTGTAGATCTGTTCT; PROBE: FAM-ACACTAGCCATCCTTACTGCGCTTCG- BHQ-1; SARS2ER: ATATTGCAGCAGTACG-CACACA. To generate a standard curve, the cDNA of SARS-CoV-2 *E* gene sgmRNA was cloned into a pCR2.1-TOPO plasmid. The copy number of sgmRNA was calculated by comparing with a standard curve obtained with serial dilutions of the standard plasmid. The detection limit of the sgmRNA was determined to be 25 copies/reaction. Values below detection limits were mathematically extrapolated based on the standard curves for graphing purpose. When graphing the results in Prism 8, values below the limit of detection were arbitrarily set to half of the LoD values.

## Histopathology analyses

Tissues (hearts, livers, spleens, duodenums, brains, lungs, kidneys, trachea, salivary gland, and nasal turbinates) were fixed in 10% neural buffered formalin overnight and then processed for paraffin embedding. The 4-$\mu$m sections were stained with hematoxylin and eosin for histopathological examinations. Images were scanned using an Aperio ImageScope.

## In-situ hybridization

To detect SARS-CoV-2 genomic RNA in FFPE tissues, in situ hybridization (ISH) was performed using the RNAscope 2.5 HD RED kit, a single plex assay method (Cat. no. 322373; Advanced Cell Diagnostics) according to the manufacturer's instructions. Briefly, Mm PPIB probe detecting peptidylprolyl isomerase B gene (housekeeping gene) (Cat. no. 313911, positive-control RNA probe), dapB probe detecting dihydrodipicolinate reductase gene from *Bacillus subtilis* strain SMY (a soil bacterium) (Cat. no. 310043, negative-control RNA probe), and V-SARS-Cov-2-S (Cat. no. 854841) targeting SARS-CoV-2 positive-sense (genomic) RNA. Tissue sections were deparaffinized with xylene, underwent a series of ethanol washes and peroxidase blocking, and were then heated in kit-provided antigen retrieval buffer and digested by kit-provided proteinase. Sections were exposed to ISH target probes and incubated at 40°C in a hybridization oven for 2 h. After rinsing, ISH signal was amplified using kit-provided pre-amplifier and amplifier conjugated to alkaline phosphatase and incubated with a fast-red substrate solution for 10 min at room temperature. Sections were then stained with 50% hematoxylin solution followed by 0.02% ammonium water treatment, dried in a 60°C dry oven, mounted, and stored at 4°C until image analysis.

# Supplementary Information

# Acknowledgements

We thank Drs. Carolyn Wilson, Jerry Weir, and Robin Levis for their support of this study and Drs. Marian Major and Surender Khurana for critical reading of the manuscript. We thank Ms. Katherine Shea and Dr. Hongquan Wan for assistance with pathology slides and Mr. Andy LaClair and Ms. Michele Howard for assistance with working in the BSL-3 laboratory. We are particularly grateful to US Food and Drug Administration White Oak Division of Veterinary Services staff and contractors who assisted in the hamster studies. The following reagent was deposited by the Centers for Diseases Control and Prevention and obtained through Biodefense and Emerging Infections Research Resources Repository, NIAID, NIH: SARS-Related Coronavirus 2, Isolate USA-WA1/2020, NR-52281. The work described in this manuscript was supported by US FDA intramural grant funds. The funders had no role in study design, data collection and analysis, decision to publish, or preparation of the manuscript. The content of this publication does not necessarily reflect the views or policies of the Department of Health and Human Services, nor does mention of trade names, commercial products, or organizations imply endorsement by the US Government.

## Author Contributions

P Selvaraj: data curation, formal analysis, and methodology.
CZ Lien: conceptualization, investigation, methodology, and project administration.
S Liu: data curation and formal analysis.
CB Stauft: data curation.
IA Nunez: data curation.
M Hernandez: methodology.
E Nimako: investigation and methodology.
MA Ortega: investigation and methodology.
MF Starost: investigation and methodology.
JU Dennis: investigation and methodology.
TT Wang: conceptualization, data curation, and formal analysis.

## Conflict of Interest Statement

The authors declare that they have no conflict of interest.

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
