## [Reviewer comments · Life Science Alliance]

Life Science Alliance

SARS-CoV-2 infection induces protective immunity and limits transmission in Syrian hamsters

Prabhuanand Selvaraj, Christopher Lien, Shufeng Liu, Charles Stauff, Ivette Nunez, Mario Hernandez, Eric Nimako, Mario Ortega, Matthew Starost, John Dennis, and Tony Wang
DOI: <https://doi.org/10.26508/lsa.202000886>

Corresponding author(s): Tony Wang, US FDA

Review Timeline:

Submission Date:	2020-08-19
Editorial Decision:	2020-10-13
Appeal Received:	2020-10-26
Editorial Decision:	2020-11-17
Revision Received:	2020-12-14
Editorial Decision:	2021-01-19
Revision Received:	2021-01-20
Accepted:	2021-01-21

Scientific Editor: Shachi Bhatt

Transaction Report:

October 13, 2020

Re: Life Science Alliance manuscript #LSA-2020-00886-T

Dr. Tony Wang
US FDA
Silver Spring

Dear Dr. Wang,

Thank you for submitting your manuscript entitled "SARS-CoV-2 infection induces protective immunity in the lung and limits transmission in Syrian hamsters". The manuscript has been evaluated by expert reviewers, whose reports are appended below. We apologize for this delay in getting back to you.

Unfortunately, after an assessment of the reviewer feedback, our editorial decision is against publication in Life Science Alliance (LSA). Given the significant technical and conceptual concerns raised by the reviewers, we can not move forward with the current version of the manuscript. However, the topic remains of interest to Life Science Alliance, and we would be willing to re-consider the paper if all of the concerns raised by the referees are adequately addressed. If you wish to submit a revised manuscript to LSA, we encourage you to appeal through our submission system, along with a point-by-point rebuttal and a revised manuscript, when ready. Please note that priority and novelty would be reassessed at resubmission

Although your manuscript is intriguing, I feel that the points raised by the reviewers are more substantial than can be addressed in a typical revision period. If you wish to expedite publication of the current data, it may be best to pursue publication at another journal.

Regardless of how you choose to proceed, we hope that the comments below will prove constructive as your work progresses. We would be happy to discuss the reviewer comments further once you've had a chance to consider the points raised in this letter.

Thank you for thinking of Life Science Alliance as an appropriate place to publish your work.

Sincerely,

Shachi Bhatt, Ph.D.
Executive Editor
Life Science Alliance
<https://www.life-science-alliance.org/>
Tweet @SciBhatt @LSAJournal

Reviewer #1 (Comments to the Authors (Required)):

Lien et al is a well written study describing a model of SARS-CoV-2 infection in young and aged Syrian hamsters. This is similar to recently published Imai et al, which showed Syrian hamsters as a potential model for Sars-CoV-2 infection. Uniquely, the authors show that rechallenged hamsters could not transmit the virus, and is severely impaired from transmitting the virus. This is especially important for public health policies and attention should be brought to it.

Major comments:

- 1) I am unclear how neutralizing antibody titers were measured (Figure 1O). Was this by PRNT? In this figure, was it only for the animals infected with 100TCID50? How about PRNT for animals infected with other titers? Neutralization assay is missing from the methods.
- 2) "Contact hamster developed viral pneumonia and seroconverted despite no observed weight loss" - Was there any viral load in the lung as measured by plaque assay or TCID50?
- 3) Recent studies by Song et al showed neuroinvasion of Sars-CoV-2 in human brain. Similarly, Imai et al found virus in the brains of infected hamsters. Did the authors find virus in the brains of the young hamsters (Figure 1E)? More importantly, did the authors check for brain viral loads in older hamsters that succumbed to the lethal infection? This could potentially lead to a model to understand why some patients present with viral neuroinvasion.
- 4) Clinically, patients with severe disease have been shown to present with a cytokine storm. Was there any indication of a cytokine storm (either by luminex or transcriptomics) in the infected animals?
- 5) Viral RNA that was detected in the lungs (Figure 3E) is unlikely to be from the input challenge virus 7 days post infection. Viruses would have been cleared or degraded at this point. 4 logs of virus is quite high, can the authors comment further on this?
- 6) All viral loads in the manuscript were measured by RT-PCR. To show that these are infectious virions, can the authors provide plaque titers or TCID50 titers?
- 7) Recent studies have indicated an important role of T cells in providing protection against infection, did the authors measure T cell responses in the hamsters in the re-infection model? Perhaps the authors can comment on this in the discussion.

Minor Comments:

- 1) The manuscript lacked page or lined numbers which makes comments challenging. Could the authors add this in during the revision?
- 2) It would be helpful if the authors could use arrows to indicate on their H&E histology slides immune infiltrates, necrosis, loss of cilia or edema that is observed. Zoomed in images of the pathology would further aid any readers to interpret the images.
- 3) Dot plots in the figures have points below the threshold. Visually, it might be easier if the authors floor the negative values to baseline.
- 4) With respect to the abstract and discussion, another lethal infection of an immunocompetent mouse model was uploaded to bioarchive (de alwis et al). The authors can instead claim that they are the first lethal hamster model?

Reviewer #2 (Comments to the Authors (Required)):

In this study, Lien and colleagues developed a Syrian hamster animal model for COVID-19 research studies. In agreement with previously reported studies, the authors show that aged hamsters experienced more pronounced signs of disease and more consistent weight loss than young hamsters. Infected hamsters were protected from a second challenge 28 days after the primary infection, and did not infect naïve animals.

This work is mostly a confirmatory study as other groups have previously reported similar findings in greater detail. Some of the data provided are very preliminary and are not strong enough to draw significant conclusion.

Specific Issues:

1. Viral titers should be determined by plaque assay or TCID₅₀ and not only by qRT-PCR.

2. IHC analysis for detection of viral antigens would be a nice addition to the histopathology analysis.

3. In figure 2 the authors compare one or two animals/group. Increasing the number of animals in the study, and including a more detailed analysis of virus replication in different organs would help strengthening their conclusions. A full necropsy evaluation of the deceased hamster should have been performed to understand whether this death was related to SARS-CoV-2 infection or not.

4. The experiments described in Figure 3 and 4 lack important controls. Figure 3 doesn't include a mock infection control for the primary infection. Figure 4 doesn't include a positive control for transmission.

Dear Dr. Bhatt,

The authors of manuscript #LSA-2020-00886-T have requested an appeal. Their comments are below.

Dear Editor,

We are submitting a revised manuscript titled "SARS-CoV-2 infection induces protective immunity in the lung and limits transmission in Syrian hamsters" (LSA-2020-00886-T) for consideration of publication in Life Science Alliance.

The initial review, while recognizing the topic of the study is of interest to Life Science Alliance, raised a series of concerns. As summarized by the editor, "Although your manuscript is intriguing, I feel that the points raised by the reviewers are more substantial than can be addressed in a typical revision period". We are pleased to report that we have accumulated more data while waiting for the review comments. In the revised manuscript, we have included the following results to strengthen our conclusions:

1. HE slides of brain and olfactory bulb from uninfected and infected hamsters (revised Fig. S1)
2. Detection of viral genomic RNA in the lung of infected hamsters either through contact transmission or direct inoculation (revised Fig. S2I.J and Fig. 2G-I).
3. Additional lethality in aged hamsters (revised Fig. 2J).
4. Virus titers in nasal washes (measured by TCID50 assay) were added to Fig. 3E.
5. Positive control for transmission-mediated infection (revised Fig. 4C)

A point-by-point response to the referee comments is also attached to this submission.

We look forward to a favorable review on this very important study. Thank you!

Tony

You can accept or decline this request from the manuscript using the following link:

<https://lsa.msubmit.net/cgi-bin/main.plex?el=A6Na6Wy4A7QqP2F4A9ftded3Z0D7hJC4jBde8cCID1AZ>

Sincerely,

Editorial Staff

MS: LSA-2020-00886-T

Dr. Tony Wang
US FDA
Silver Spring

Dear Dr. Wang,

We have considered your appeal for our decision on "SARS-CoV-2 infection induces protective immunity in the lung and limits transmission in Syrian hamsters". After assessing the revised manuscript, point-by-point rebuttal and your appeal letter, the academic editors and I are pleased to let you know that we have decided to send your manuscript for external review.

Please use the following link to submit your revised manuscript:

<https://lsa.msubmit.net/cgi-bin/main.plex?el=A4Na6Wy3A3Cjtu4I1B9ftdYS3aDNv9d34WwZ6boTBsPgZ>

We will let you know when the reviews have been received and a decision has been made.

Yours sincerely,

Shachi Bhatt, Ph.D.
Executive Editor
Life Science Alliance
<https://www.lsjournal.org/>
Tweet @SciBhatt @LSAjournal

The revision contains the following additional new results to strengthen our conclusions:

1. HE slides of brain and olfactory bulb from uninfected and infected hamsters (revised **Fig. S1M-R**).
2. Detection of viral genomic RNA in the lung of infected hamsters either through contact transmission or direct inoculation (revised **Fig. S2I-J and Fig. 2G-I**).
3. Additional lethality in aged hamsters (revised **Fig. 2J**).
4. Virus titers in nasal washes (measured by TCID₅₀ assay) were added to **Fig. 3E**.
5. Positive control for transmission-mediated infection (revised **Fig. 4C**)

A point-by-point response to the referee comments is provided as below (original review comments in blue and our response in black):

Reviewer: 1

1- I am unclear how neutralizing antibody titers were measured (Figure 1O). Was this by PRNT? In this figure, was it only for the animals infected with 100TCID₅₀? How about PRNT for animals infected with other titers? Neutralization assay is missing from the methods.

Our response: The neutralizing antibody titers (nABs) shown in Figure 1O were determined using a SARS-CoV-2 pseudovirus neutralization assay, which had been described in the *Materials and Methods* section. This is an assay that we have qualified here at FDA with good correlation with titers obtained from the standard PRNT. The nAB titers shown in Figure 1O included nine hamsters that were challenged with 10⁵ TCID₅₀ (shown in black solid circles) as well as those with 100 and 10 TCID₅₀ in blue and green circles, respectively.

2- "Contact hamster developed viral pneumonia and seroconverted despite no observed weight loss" - Was there any viral load in the lung as measured by plaque assay or TCID₅₀?

Our response: We did not save the lung tissue of this particular "contact hamster" for plaque assay. However, we did perform RNAscope to show the presence of viral genomic RNA in the infected lung (**Fig. S2I-J**).

3- Recent studies by Song et al showed neuroinvasion of Sars-CoV-2 in human brain. Similarly, Imai et al found virus in the brains of infected hamsters. Did the authors find virus in the brains of the young hamsters (Figure 1E)? More importantly, did the authors check for brain viral loads in older hamsters that succumbed to the lethal infection? This could potentially lead to a model to understand why some patients present with viral neuroinvasion.

Our response: We are aware of the work by Song et al and Imai et al. Multiple reports have shown that SARS-CoV-2 infection of K18-hACE2 mice is dose-dependently fatal from 5 dpi (Golden, J. W. et al., Perlman, S., and Oladunni, F. S. et al.). It must be pointed out, however, SARS-CoV-2 neuroinvasion, but not respiratory infection, is associated with mortality in this transgenic mouse model. By contrast, patients who succumb to SARS-CoV-2 infection typically die of acute respiratory distress syndrome (ARDS). To the best of our knowledge, most groups who work with the hamster model have not been

able to cultivate live infectious virus from the brains of hamsters that are challenged with SARS-CoV-2. We have repetitively examined brains, including the olfactory bulbs of infected hamsters and could not find any significant lesion. Having said that, we cannot rule out the possibility that a transient low degree infection may occur in the olfactory bulb or even in the brain. Nonetheless, HE images and RNAscope images from uninfected and infected hamster brains can be found in the Fig. 2G-I and Fig. S1M-R.

4- Clinically, patients with severe disease have been shown to present with a cytokine storm. Was there any indication of a cytokine storm (either by luminex or transcriptomics) in the infected animals?

Our response: This is an excellent question that is worth further investigation. One of the caveats working with Syrian hamsters is the relative lack of reagents for conducting immunological assays. To the best of our knowledge, there is no luminex assay available for hamsters to directly measure cytokines. Transcriptomics analyses of infected hamster tissues do show the upregulation of cytokine genes (Benjamin R. tenOever et al), although it is unclear whether infection induces a cytokine storm. With the aged hamster model in hand, we are exploring several techniques to investigate along this line and hopefully will be able to provide an update in the future.

5- Viral RNA that was detected in the lungs (Figure 3E) is unlikely to be from the input challenge virus 7 days post infection. Viruses would have been cleared or degraded at this point. 4 logs of virus is quite high, can the authors comment further on this?

Our response: we thank the reviewer for the comment. We typically see 7-10 logs of virus in the infected lungs, so 3-4 logs of virus (vRNA titers) was indeed quite low given that the genomic RNA/PFU ratio of SARS-CoV-2 is about 1000-10,000 as reported by Plante et al. In other words, 3-4 logs of virus (in vRNA titers) may have only 1-10 infectious virus particles. Nonetheless, we deleted the phrase “likely be from the input challenge virus” in the revision.

6- All viral loads in the manuscript were measured by RT-PCR. To show that these are infectious virions, can the authors provide plaque titers or TCID₅₀ titers?

Our response: We have updated Figure 3E to include TCID₅₀ titers in nasal washes.

7- Recent studies have indicated an important role of T cells in providing protection against infection, did the authors measure T cell responses in the hamsters in the re-infection model? Perhaps the authors can comment on this in the discussion.

Our response: we have not measured T cell responses in hamsters, primarily because of lacking reagents. Nonetheless, we have attempted to block T cell response using a commercial antibody and have seen extended period of infection in hamsters. Jay Hooper’s group at USAMRIID reported that Cyclophosphamide (CyP) immunosuppressed or RAG2 knockout (KO) hamsters developed clinical signs of disease that were more severe than in immunocompetent hamsters upon SARS-CoV-2 challenge, suggesting that functional B and/or T cells are likely to play an important role for the clearance of SARS-CoV-2 and in protection from acute disease.

Minor comments

1) The manuscript lacked page or lined numbers which makes comments challenging. Could the authors add this in during the revision?

Our response: We thank the reviewer for the advice and have included page and line numbers to the revised manuscript.

2) It would be helpful if the authors could use arrows to indicate on their H&E histology slides immune infiltrates, necrosis, loss of cilia or edema that is observed. Zoomed in images of the pathology would further aid any readers to interpret the images.

Our response: We thank the reviewer for the advice and have added indicators for histopathology changes and a closeup image (**Fig. S1L**). We would also like to kindly remind that there are several hundred HE slides related to this study and many of them cannot be included in the manuscript due to space constraint. Nonetheless, all HE images that are presented in the manuscript are captured in high resolution, meaning that the readers should be able to Zoom in easily and see details as they wish.

3) Dot plots in the figures have points below the threshold. Visually, it might be easier if the authors floor the negative values to baseline.

Our response: We amended all figures as the reviewer recommended.

4) With respect to the abstract and discussion, another lethal infection of an immunocompetent mouse model was uploaded to bioarchive (de alwis et al). The authors can instead claim that they are the first lethal hamster model?

Our response: De alwis et al were using the hACE2 transgenic mouse model, which develops lethality after SARS-CoV-2 infection due to neuroinvasion, but not respiratory infection. This finding has been documented by multiple groups. To distinguish our model from the transgenic mouse model, we have now amended the relevant text to read “the first lethal model using genetically unmodified laboratory animals”.

Reviewer: 2

Comments to the Author

In this study, Lien and colleagues developed a Syrian hamster animal model for COVID-19 research studies. In agreement with previously reported studies, the authors show that aged hamsters experienced more pronounced signs of disease and more consistent weight loss than young hamsters. Infected hamsters were protected from a second challenge 28 days after the primary infection and did not infect naïve animals.

This work is mostly a confirmatory study as other groups have previously reported similar findings in greater detail. Some of the data provided are very preliminary and are not strong enough to draw significant conclusion.

Our response: We were among the earliest groups who started the Syrian hamster model for SARS-CoV-2. Most results included in the manuscript were first presented to a World Health Organization working group on July 9, 2020. *The intention of our study is not to confirm what other have reported.* The novelty of this study resides in twofold: 1) we have established a lethal animal model for COVID-19. This is very significant because the only other animal model that sometimes shows lethality is the K18 hACE2 transgenic mouse model. The cause of death in a fraction of SARS-CoV-2 challenged hACE2 transgenic mice is neuroinvasion, which differs from severe COVID-19 cases where patients

mainly die of ARDS. Using aged hamsters without any genetic modification, we consistently observed lethality. This aged hamster model not only allows investigation of the pathogenesis of severe and even fatal COVID-19 cases, but also represents a superior platform for evaluating efficacies of vaccines and antivirals.

2) This study attempted to address one of most important questions regarding SARS-CoV-2, i.e., prior infections will likely protect from a secondary SARS-CoV-2 infection, but will it prevent transmission? Illustrated in the figure below, when a convalescent individual (**Subject A**) becomes re-exposed to a virus carrier (the **Spreader**), there are likely two scenarios: 1, the immunity that **Subject A** has acquired from prior infection not only protects him/her from getting pneumonia, but also prevents transmission to

other naïve individuals because **Subject A** sheds no or very little virus. Under such a scenario, the spread of the virus through **Subject A** is virtually stopped. 2, **Subject A** is protected from developing severe diseases but continues to shed virus because the acquired immunity is not strong enough to curtail the replication of virus in the nose or upper respiratory tract. In this case, **Subject A** serves as a virus spreader.

Our results clearly demonstrated that scenario #1 can be achieved in the hamster model. Ongoing research in the laboratory is to examine whether a vaccine would accomplish the same. Such information is critical for decision-making of public health policies and evaluating efficacy of vaccine candidates. There have been no published studies that directly address this important question.

Lastly, we provided additional data to strengthen the conclusions. We apologize that perhaps because we have worked with the hamsters for so long and hence have taken many findings for granted. For example, we have done the transmission study many times and it works consistently, however, this information is not obvious to the reviewer. Therefore, we may have skipped some controls that the reviewer deems essential.

Specific Issues:

1- Viral titers should be determined by plaque assay or TCID₅₀ and not only by qRT-PCR.

Our response: We have updated Figure 3E to include TCID₅₀ titers in nasal washes.

2 - IHC analysis for detection of viral antigens would be a nice addition to the histopathology analysis.

Our response: In the revised manuscript, we have included RNAscope data to demonstrate the infection of lungs.

3 - In figure 2 the authors compare one or two animals/group. Increasing the number of animals in the study and including a more detailed analysis of virus replication in different organs would help strengthening their conclusions. A full necropsy evaluation of the deceased hamster should have been performed to understand whether this death was related to SARS-CoV-2 infection or not.

Our response: To clarify, in the experiment shown in Figure 2, we had 20 hamsters for the 10⁵ TCID₅₀, 2 for 10⁴, 1000, 100 and 10 TCID₅₀ group. Not all hamsters were sacrificed on the same day or at the end of the study, therefore we only plotted graphs using matched samples. Therefore, the readers see fewer data points in Figure 2. Furthermore, we have challenged hamsters using different doses in several

different experiments and obtained similar observations, but because these experiments were conducted on different dates, we did not pool results from those hamsters in the graphs. Nonetheless, the conclusions remain the same.

Regarding the cause of the death of the deceased hamster (**Fig. 2**), a full necropsy indeed was performed. Lesions are not observed in the pancreas, small intestine, large intestine, brain. The animal had severe, diffuse bronchointerstitial pneumonia along with myocardial disease and thrombus formation in the left atrium, leading to death of this animal. RNAscope images are now provided to show the presence of viral genomic RNA in the infected hamster lung but not in the brain (**Fig. 2G-I**).

4 –The experiments described in Figure 3 and 4 lack important controls. Figure 3 doesn't include a mock infection control for the primary infection. Figure 4 doesn't include a positive control for transmission.

Our response: We started working with the hamster model as early as March 2020 and the experimental procedures have been optimized to ensure reproducibility. For example, the same lot of virus has been used up to date for inoculation; inoculated hamsters always develop pneumonia with measurable viral loads in the lung ranging between 10^7 and 10^9 copies/0.1g; transmission-mediated infection of hamsters always occurs under the current protocol. Such information, although seems routine to us, were not mentioned in the initial submission which might have prompted the reviewer to think that the experiments described in Figure 3 and 4 lack important controls.

Particularly, hamsters in Figure 3 were previously infected at the same time with those hamsters shown in Figure 1. The re-infection was done under the same procedure using the same lot of virus. Also, at the time when we were performing the experiment reported in Figure 3, another project that was being simultaneously conducted in the laboratory did include hamsters that could serve as mock infection control. These hamsters readily developed pneumonia upon challenge and showed much higher viral loads in the lung. Because the results are intended for another project, we could not include them as a mock infection control group as the reviewer would like to see. Nonetheless, the results and conclusions of Figure 3 are well supported.

Same thing could be said to Figure 4. The procedure has been established to provide transmission at 100% success rate. To satisfy the reviewer, we now included data points of two contact hamsters that were exposed infected hamsters as positive controls for transmission (revised **Fig. 4C**).

In summary, we believe the revised manuscript has been improved significantly and hope these changes will warrant a positive decision.

Sincerely,

Tony Wang, PhD
Principal Investigator
Laboratory of Vector-borne Viral Diseases

Division of Viral Products|CBER
U.S. Food and Drug Administration
10903 New Hampshire Avenue

Silver Spring, MD 20993
Office: Bldg. 52/72, Rm 5336
Lab: Bldg. 52/72, Rm 5353
Office Phone: (240) 402 – 1956

January 19, 2021

RE: Life Science Alliance Manuscript #LSA-2020-00886-TR-A

Dr. Tony Wang
US FDA
10993 New Hampshire Avenue
Silver Spring 20993

Dear Dr. Wang,

Thank you for submitting your revised manuscript entitled "SARS-CoV-2 infection induces protection in the lung and limits transmission in Syrian hamsters". We would be happy to publish your paper in Life Science Alliance pending final revisions necessary to meet the minor text edits requested by Rev 1 and our formatting guidelines.

We apologize for this delay in getting back to you, one of the reviewers took longer than expected to send us their re-review.

Along with the points listed below, please also attend to the following:

- please make sure that the author list in our system matches with the author list in your manuscript and that each contributing author is entered in our system
- please add ORCID ID for corresponding author-you should have received instructions on how to do so
- please add a Summary Blurb/Alternate Abstract & a Category for your research in our system
- please add your supplementary figure legends to your main manuscript text and upload your supplementary figures as single files
- please use the [10 author names, et al.] format in your references (i.e. limit the author names to the first 10)
- please add a callout for Figure S2A & S2C-F in your main manuscript text
- please make sure that the scale bars in Figure 1G-N, 2E-I, 3G, 4E, S1 and S2 are more visible. In their current form they are hard to notice and read.

A. FINAL FILES:

B. MANUSCRIPT ORGANIZATION AND FORMATTING:

Sincerely,

Shachi Bhatt, Ph.D.

Executive Editor
Life Science Alliance
<https://www.lsjournal.org/>
Tweet @SciBhatt @LSAJournal

Reviewer #1 (Comments to the Authors (Required)):

The authors have address most of my concerns, however, there are a few minor points

With response to point 6, the authors only included TCID50 data for Figure 3. How about for Figure 1 or 2? If TCID50 was not performed, perhaps they could explain why they chose to perform TCID50 for some experiments, sgRNA for some experiments and vRNA for others for consistency.

With reference to Figure 1B, Day 7 has 5 points with n=3 animals? Could the authors check on these data points.

More importantly, I agree with Reviewer 2 that Figure 3 requires a mock infection control group that the authors should try to either include, or cite a previously published paper by their group with the exact same method.

Reviewer #2 (Comments to the Authors (Required)):

The authors have satisfactorily responded to all my questions and made the necessary changes to the manuscript. The manuscript looks ready for publication.

January 21, 2021

RE: Life Science Alliance Manuscript #LSA-2020-00886-TRR

Dr. Tony Wang
US FDA
10993 New Hampshire Avenue
Silver Spring 20993

Dear Dr. Wang,

Thank you for submitting your Research Article entitled "SARS-CoV-2 infection induces protective immunity and limits transmission in Syrian hamsters". It is a pleasure to let you know that your manuscript is now accepted for publication in Life Science Alliance. Congratulations on this interesting work.

DISTRIBUTION OF MATERIALS:

Again, congratulations on a very nice paper. I hope you found the review process to be constructive and are pleased with how the manuscript was handled editorially. We look forward to future exciting submissions from your lab.

Sincerely,

Shachi Bhatt, Ph.D.
Executive Editor
Life Science Alliance
<https://www.lsjournal.org/>
Tweet @SciBhatt @LSAJournal